# Diaper dermatitis and associated factors among children aged 0–24 months in low- and middle-income countries: A systematic review protocol

**Atoma Negera[1]\*, Midekso Sento[2], Debela Dereje[1], Gamachis Firdisa[1], Samuel Negera[3], Geleta Nenko[4], Tadesse Sime[1], Merga Keba[5]**

1 Nursing Department, College of Health Science, Mattu University, Mattu Town, Oromia Regional State, Ethiopia, 2 Biomedical Department, Anatomy Course Unit, Adama Hospital Medical College, Adama Town, Oromia Regional State, Ethiopia, 3 Medical Laboratory Department, College of Health Science, Oda Bultum University, Chiro Town, Oromia Regional State, Ethiopia, 4 Health Informatics Department, College of Health Science, Mattu University, Mattu Town, Oromia Regional State, Ethiopia, 5 PHEM Officer, Horo Guduru Wallaga Zone, Oromia Regional State, Ethiopia

\* atomanegera@gmail.com

## Abstract

### Background

Diaper dermatitis (DD, sometimes known as diaper rash or napkin dermatitis is one of the most prevalent skin infections that occur in the area covered by the diaper. Although diaper dermatitis can be seen in any patient wearing diapers, it is reported to be more common in infants aged 9–12 months. Approximately, 70% of infants and young children suffer from diaper dermatitis at some point during their diaper wearing years and up to 25% of children seek healthcare due to this case.

### Methods and analysis: Methods and analysis

A comprehensive literature search will be gathered from electronic databases such as PubMed, Embase, Hinari, Cochrane Library, African Journals Online (AJOL), and Google Scholar. The protocol followed the Preferred Reporting Items for Systematic Reviews and Meta-analyses for Protocol (PRISMA-P) guideline. All studies conducted in low- and middle-income countries (LMICs) will be included regardless of their study design as long as these studies report the magnitude of the problem under study. Joanna Briggs Institute's (JBI) appraisal checklist will be used to assess the quality of individual studies. Heterogeneity will be checked using Cochrane Q test statistics and $I^2$ test statistics, and a random-effects model will be employed to estimate the pooled prevalence of DD and its associated factors.

### Ethics and dissemination of research

Ethics committee approval or written informed consent will not be required to conduct this review, and meta-analysis for the review will be entirely based on published data. The result of this study will be submitted to a peer-reviewed journal, and it will also be presented at relevant research conferences.

**Data Availability Statement:** Research data will be made publicly available when the study is completed and published.

**Funding:** The author(s) received no specific funding for this work.

**Competing interests:** The authors have declared that no competing interests exist.

## Results

The present study will estimate the pooled prevalence of diaper dermatitis and their associated factors in low- and middle-income countries.

## Trial registration

**PROSPERO registration number:** CRD42024578550.

## Introduction

Diaper dermatitis (DD), sometimes known as diaper rash or napkin dermatitis, is one of the most prevalent skin infections that occur in the area covered by the diaper [1,2]. Although diaper dermatitis can be seen in any patient wearing diapers, it is reported to be more common in infants between the ages of 9–12 months who are starting to wean off of breast milk and consume solid meals [3–6]. The most frequent cause of diaper dermatitis is irritant contact dermatitis. These conditions can range in severity from minor (persistent redness) to major (destruction of the epidermis). It's typical sign is a red, itchy, and sometimes ulcerated rash in the diaper area and may include the presence of papules and pustules [7,8].

The primary diaper sites include the perineum, peri-genital area, perianal area, genital region, buttocks, and groin and occur most often as a reaction to increased moisture (skin overhydration) due to prolonged contact with urine, feces, or retained soap or detergent [9,10]. Major causes that impair the integrity of the skin are irritants in urine and feces, over-hydration of the epidermal stratum and corneum, elevated skin pH, and friction against the skin [10–12]. The longer the feces stay in contact with the skin, the higher the risk of DD. The urea in urine, which is converted to ammonia by fecal bacteria with urease activity, and the proteolytic enzymes present in feces are believed to cause irritant dermatitis [5].

Now days, disposable diapers come with a lot of features that are advantageous for newborns, particularly in terms of skin protection, lowering the incidence of diaper dermatitis through effective absorbency, and lowering the skin area pH [13–15]. Globally, diapers are made of a variety of materials, with disposables being the most often used. Disposable diapers are single-use products made of absorbent gel materials (AGM), which absorb liquid and runny feces. Cloth diapers are made of layers of material, such as microfiber, cotton, hemp, or bamboo, and they are reusable after washing [10].

The prevalence of DD varies by country and study year. Globally, it is estimated that about 70% of infants and young children experience diaper dermatitis at some point during their diaper-wearing years, and up to 25% of children seek medical care as a result of this case [6,16–18]. Evidence from Asia and Africa reported the case as 67.3% in India [19], 62.5% in Ethiopia [20], 50.9% in Iran [21], 38.9% in Nigeria [22], 36.1% in Thailand [23], 33.6 in Bangladesh [24], 27.4% in Ghana [25], 27.3% in Kenya [26], and 18.4% in Cameron [27]. Infants and younger children in LMIC change diapers approximately 8 to 10 times per day and wear an average of more than 3,500 diapers before they are toilet trained [28]. Baby diapering is a common practice, however in developing countries, there is little information available about it. According to a study from sub-Saharan Africa, the consumption of disposable diapers varies from 59 children per year in Nigeria to 433 children per year in Kenya [29].

In low- and middle-income countries (LMIC), where there is a high birth rate, inadequate water supply, scarcity of disposable diapers and wipes, lack of access to healthcare, higher rates

of comorbid diseases, and low awareness of mothers regarding the condition and its management [24,30–32], we believe that the actual prevalence of DD may be higher than the reported. The above evidences show that, the subject needs to be given more emphasis. Thus, by estimating its pooled prevalence and identifying associated factors, this study will contribute to the prevention and management of DD. Furthermore, the findings of the current study may encourage researchers to give more emphasis to conducting further studies on the topic by using this study as baseline evidence.

To avoid duplication, a preliminary search of similar reviews or review protocols was conducted in the databases of Google Scholar, MEDLINE, and The International Prospective Register of Systematic Reviews (PROSPERO). To our knowledge, no review or planned review was identified on the subject that has been published within the last decade, despite the fact that a significant percentage of children suffer from it and there is little information available on diaper dermatitis in the LMICs. Therefore, this review aims to answer the following research questions: 1) What is the pooled prevalence of diaper dermatitis among children aged 0–24 months in low- and middle-income countries? 2) What are the factors associated with diaper dermatitis among children aged 0–24 months in low- and middle-income countries?

## Methods

### Registration and reporting

The review was registered in the International Prospective Register of Systematic Reviews (PROSPERO) with registration number CRD42024578550. This protocol followed the Preferred Reporting Items for Systematic Reviews and Meta-Analysis for Protocol (PRISMA-P) checklist [33]. The review will be commenced between September 20 and October 20, 2024, and necessary amendments will be published along with the results of the systematic review and meta-analysis.

### Information sources

An electronic database search for published articles and gray literature will be done from the following databases: PubMed, Embase (Ovid), Hinari, Cochrane Library, African Journals Online (AJOL), Google Scholar, and other maternal, neonatal, and pediatrics databases. Additional searches from references to identified literature will be performed. All research articles containing information on the prevalence of symptomatic diaper dermatitis/rashes or at least one associated factor of diaper dermatitis/rashes in children less than 24 months will be retrieved without a time limit.

### Search strategy

The search strategy included a combination of subject terms and free text terms combined with Boolean operators ('OR', and 'AND'). The Medical Subject Headings (MeSH) terms included "diaper", OR "napkin", OR "nappy" AND "dermatitis", OR "rash", OR "irritation", OR "papule", OR "pustule" AND "neonate", OR "infant", OR "toddler", OR "children", OR "pediatrics", OR "kids", OR "babies", AND "income", OR "developing countries", OR "resource-limit" OR "resource-poor", OR "low-income", OR "lower-middle-income", OR "middle-upper income countries".

### Eligibility criteria

**Inclusion criteria. Population:** Studies involving children aged 0–24 months who visited health facility in low- and middle-income countries.

**Exposure**: Diaper dermatitis among children aged 0–24 months in low- and middle-income countries.

**Study design:** Observational studies (i.e, cross-sectional, case-control, retrospective and prospective cohort studies and national survey and surveillance reports) will be considered for this review.

**Study area:** Only studies conducted in LMICs and reported the prevalence of DDs will be included. However, if multiple publications are generated from the same data with the same outcome, only the most relevant publications will be considered.

**Publication language:** Any research article regardless of published language. If the study was reported in languages other than English, online translation software will be used to extract the necessary data.

**Search date:** All research articles accessed from 10 to 30 November 2024 without time limits will be included.

**Exclusion criteria.**   Editorials, opinion articles, letters, narrative or systematic reviews, brief communications, conference abstracts, and posters will be excluded. Studies that did not assess the magnitude of diaper dermatitis/rashes (studies that evaluate treatments for children with DD, knowledge of DD, the quality of life of children with DD, etc.) will be excluded.

## Study selection and data extraction

All identified articles from databases will be exported to the EndNote Software [34] and initially screened by title and abstract. The full text of those articles satisfying inclusion criteria by title and abstract will be reviewed the full articles. A Microsoft Excel spreadsheet and standardised data extraction form will be developed based on Cochrane Good Practice Data Extraction Template [35]. Abstracts of the relevant full texts will be assessed for eligibility by two reviewers (AN and MS) independently. Full-text articles for the selected titles will be further reviewed independently by these reviewers. Disagreements will be resolved by consensus where possible or by a third reviewer (GF) as needed. Two of the authors (DD and SN) will extract data independently using a customized data extraction form. All selected papers will be finally cross-checked by two other authors (GN and TS). Any disagreement on a particular paper will be resolved by discussion before inclusion in the study. Data extracted will include the author(s), journal, year of publication, type of the study, study design, country of origin, rationale of the study, study population, sample size, outcomes of the study, key findings related to the review, limitations, and recommendations. Author(s) will be contacted for those studies in which raw data were missing or unclear.

## Risk-of-bias and quality assessment

A risk-of-bias tool will be assessed using the Joanna Briggs Institute (JBI) appraisal checklist developed explicitly for systematic reviews of prevalence studies [36]. Two review authors (AN and MK) will assess the risk of bias independently and assign each study a JBI score ranging from 1 to 10, with higher scores indicating higher quality; inconsistencies will be identified and resolved through discussion involving a third author where necessary.

## Data synthesise

The extracted data will be exported to Stata version 15 statistical software for analysis [37]. The pooled prevalence rates will be calculated from raw proportions reported in the included studies. Summary statistics of individual studies will be presented in a table and graph. Cochrane's Q statistic, $I^2$, and p-value will be used to check the heterogeneity of the study's outcomes. $I^2$ of 25%, 50%, and 75% will be used as an indicator for low, moderate, and high heterogeneity,

respectively [38]; forest plots will be used to visualize heterogeneity. A statistical significance result from the Egger and Begg tests will also be used as an indicator of publication bias. If there is moderate to high heterogeneity, random effect meta-analysis will be employed. Meta-regression will be used to identify the source of heterogeneity. A statistically significant result from meta-regression will be declared as a source of heterogeneity. Sub-group analysis will be conducted between the economic level of countries (low-income vs. middle-income countries), sample size of the study (large vs. small sample size), sampling techniques used (random vs. convenience), and publication year (2010 to recent years vs. years before 2010).

## Operational definition

**Absorbent Gel Materials (AGM):** are materials with a high capacity to absorb and hold large amounts of water and aqueous solutions. This has made them ideal for baby diapers and other products [39].

**Income classification:** to determine the economic class of a country, we use the World Bank's income classification system. Based on Gross National Income (GNI) per capita, the World Bank divides the economies of the nations into four income groups: low, lower-middle, upper-middle, and high income [40].

i. **Low-income countries (LICs):** are defined as nations with a Gross National Income (GNI) per capita, of 1,145 USD or less.

ii. **Lower middle-income countries (LMICs):** are nations with a GNI per capita between 1,146 and 4,515 USD

iii. **Upper middle-income countries (UMICs):** are nations with a GNI per capita between 4,516 and 14,005 USD

iv. **High-income countries (HICs):** are nations with more than 14,005 USD GNI per capita.

## Ethics approval

Ethics committee approval or written informed consent will not be required to conduct this systematic review, and meta-analysis for the review will be entirely based on published data.

## Dissemination of the results

The result of this study will be submitted to a peer-reviewed journal, and it will also be presented at relevant research conferences.

## Patient and public involvement

Patients and/or the public were not involved in the design, conduct, reporting, or dissemination plans of this research.

## Strengths and limitations of the study

✓ This systematic review will address a global gap by providing the pooled prevalence of diaper dermatitis (DD) in LMICs.

✓ The review will identify factors associated with the diaper dermatitis that will assist us in recommending responsible bodies.

✓ This review protocol follows the Preferred Reporting Items for Systematic Review and Meta-analysis Protocols (PRISMA-P) guidelines with transparency regarding the methods and processes used.

✓ Among the limitations that could be anticipated, the studies found could have a small sample size.

✓ There is a difference in the prevalence of diaper dermatitis in the population and health institution (clinic). The study may therefore not accurately reflect the prevalence of DD at community level.

✓ The study will assess neither treatment of diaper dermatitis nor knowledge of parents on DD.

## Supporting information

**S1 Checklist.**
(DOCX)

**S2 Checklist. PRISMA-P (Preferred Reporting Items for Systematic review and Meta-Analysis Protocols) 2015 checklist: Recommended items to address in a systematic review protocol\*.**
(DOCX)

## Author Contributions

**Conceptualization:** Atoma Negera, Debela Dereje, Gamachis Firdisa.

**Data curation:** Samuel Negera, Geleta Nenko.

**Funding acquisition:** Debela Dereje, Merga Keba.

**Investigation:** Debela Dereje, Samuel Negera, Merga Keba.

**Methodology:** Atoma Negera, Midekso Sento, Samuel Negera, Tadesse Sime.

**Project administration:** Atoma Negera, Gamachis Firdisa, Geleta Nenko.

**Resources:** Geleta Nenko.

**Software:** Atoma Negera.

**Supervision:** Atoma Negera.

**Validation:** Atoma Negera, Midekso Sento, Tadesse Sime.

**Visualization:** Geleta Nenko.

**Writing – original draft:** Atoma Negera, Midekso Sento, Debela Dereje, Gamachis Firdisa, Tadesse Sime.

**Writing – review & editing:** Atoma Negera, Samuel Negera, Merga Keba.

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
