## [Decision Letter · Decision Letter 0]

6 Oct 2024

PONE-D-24-40961Diaper dermatitis and associated factors among 0-24 months children in low- and middle-income countries: A Systematic review protocolPLOS ONE

Dear Dr. Negera,

Thank you for submitting your manuscript to PLOS ONE. After careful consideration, we feel that it has merit but does not fully meet PLOS ONE’s publication criteria as it currently stands. Therefore, we invite you to submit a revised version of the manuscript that addresses the points raised during the review process.

Both reviewers are supportive but have some comments to address, particularly reviewer 2.   I hope they are useful in both correcting this paper and carrying out your full study.

We look forward to receiving your revised manuscript.

Kind regards,

Alison Parker

Academic Editor

PLOS ONE

Journal Requirements:

Reviewers' comments:

Reviewer's Responses to Questions

**Comments to the Author**

1. Does the manuscript provide a valid rationale for the proposed study, with clearly identified and justified research questions?

Reviewer #1: Yes

Reviewer #2: Partly

2. Is the protocol technically sound and planned in a manner that will lead to a meaningful outcome and allow testing the stated hypotheses?

Reviewer #1: Yes

Reviewer #2: Partly

3. Is the methodology feasible and described in sufficient detail to allow the work to be replicable?

Reviewer #1: Yes

Reviewer #2: Yes

4. Have the authors described where all data underlying the findings will be made available when the study is complete?

Reviewer #1: Yes

Reviewer #2: Yes

5. Is the manuscript presented in an intelligible fashion and written in standard English?

Reviewer #1: Yes

Reviewer #2: Yes

6. Review Comments to the Author

You may also provide optional suggestions and comments to authors that they might find helpful in planning their study.

Reviewer #1: This is an intriguing and novel submission, addressing a gap in the literature. The outcomes of the meta-analysis will be valuable in contributing to ongoing efforts to reduce diaper dermatitis. However, I recommend reconsidering the paragraph on page 5, beginning with "Diaper dermatitis...," as it does not substantially enhance the background relevant to the study and could be omitted. Additionally, there is insufficient discussion regarding low- and middle-income countries within the introduction portion of the paper. It would be beneficial to expand on the significance of these populations and clarify whether comparisons will be made to developed countries. Finally, the overarching aim of the review appears somewhat simplistic, and further clarification on how the results will guide future research in diaper dermatitis would strengthen the manuscript.

Reviewer #2: Reviewer comments:

Overall:

It was not clear to me how “low- and middle-income countries (LMIC) will be defined. What criterion will be used?

The title of the protocol states that it will be looking at Diaper Dermatitis “and associated factors”, and the protocol states a key question as “2) What are the factors associated with diaper dermatitis among children aged 0-24 months in low- and middle-income countries?” It is not clear if this includes “causes” of diaper dermatitis, sequelae of diaper dermatitis (for example, ill temperament of the child, poor sleep, pain/crying) or other factors.

There is a statement in the “Introduction” related to the prevalence of diaper dermatitis. “Although the prevalence of DD varies from country to country, it is estimated that about 50% to 65% of infants and young children will suffer from diaper dermatitis at some time and up to 25% of children seek healthcare due to this problem (6, 7, 8).” The literature is quite scattered about the percentage of babies who experience diaper dermatitis. However, data from Visscher et al (Pediatric Dermatology Vol 17, No. 52-57, 2000) , indicates that diaper dermatitis rates are ~ 70% in the first week of life which is higher than the 50-65% reported in the publication. Further, Gustin et al (Pediatric Dermatology 2021. Jul;38(4):768-774) indicated that diaper dermatitis was found in 65-83% of newborns. Carr et al. (Pediatric Dermatology 2020 Jan;37(1):130-136) also found high rates of diaper dermatitis in Germany across 4 common sites within the diaper. Given these data, it should be noted that in reality, ‘every baby will experience diaper dermatitis as some point during their diaper wearing years’.

The Introduction also states “Its typical sign is a red, itchy, and sometimes ulcerated rash in the diaper area (perineum, peri-genital area, perianal area, and the upper thigh) occurring most often as a reaction to increased moisture due to prolonged contact with urine, feces, or retained soap or detergent (14).” I would change this to read: “It’s typical sign is a red, itchy, and sometimes ulcerated rash in the diaper area and may include the presence of papules and pustules (Buckley et al, Pediatric Dermatology Vol 33 No 6 632-639, 2016; Odio MR et al Dermatology 2000; 200:238-243). The primary diaper sites include the perineum, peri-genital area, perianal area, genital region, buttocks and groin and occur most often as a reaction to increased moisture (skin overhydration) due to prolonged contact with urine, feces, or retained soap or detergent (14).”

Introduction: “Major causes that impair the integrity of the skin are irritants in urine and feces, overhydration of the epidermal stratum and corneum, excessive skin pH, and friction against the skin (14-16).” I would reword this to say “…elevated skin pH…”

Introduction: “Disposable diapers contain absorbent chemicals that are able to absorb liquids, and have to be disposed of after usage.”. I would rephrase to state “Disposable diapers contain absorbent gel materials (AGM) that are able to absorb liquid and runny feces and are single-use.”

The Search Strategy is listed as: “The search strategy included a combination of subject terms and free text terms combined with Boolean operators (‘OR’, and ‘AND’). The Medical Subject Headings (MeSH) terms included “diaper”, OR “napkin”, OR “nappy” AND “dermatitis”, OR “rash”, OR “irritation”, AND “neonate”, OR “infant”, OR “toddler”, OR “children”, OR “pediatrics”, OR “kids”, OR “babies”, AND “developing countries”, OR “resource-limit” OR “resource-poor”, OR “low-income”, OR

“lower-middle-income”, OR “middle-upper income countries”.” I would also include “papule”, “pustule”, . I believe search terms for “income” can be included but I strongly suggest the investigators define the criterion for LMIC and then include only those publications where the study was conducted in a LMIC, as a publication may not identify its sources as LMIC.

The inclusion criteria includes data collected at the “under-five clinic”, however, given the episodic, unpredictable and short-term nature of diaper dermatitis (typically 3-10 days) the incidence of diaper dermatitis that is observed at a clinic visit may not reflect the actual rate of diaper dermatitis in the population.

The “Search date” is listed as “All research articles accessed from 20 September 2024 to 10 October 2024 without time limits will be included.”. This date is in the past.

In the exclusion criteria it states, “Studies that did not assess the magnitude of diaper dermatitis/rashes (studies that evaluate treatments for children with DD, knowledge of DD, the quality of life of children with DD, etc.) will be excluded.”. If parental habits and practices are able to be obtained, this will be critical to understand the potential causes of the diaper dermatitis, as are skin products being used, bathing frequency, diet information and family history of skin disorders, including atopic dermatitis and the like.

7. PLOS authors have the option to publish the peer review history of their article (what does this mean?). If published, this will include your full peer review and any attached files.

Reviewer #1: No

Reviewer #2: No

---

## [Author Response · Author response to Decision Letter 0]

10 Oct 2024

Dear Editor, 

Thank you for the opportunity to submit a revised version of the manuscript “Diaper dermatitis and associated factors among children aged 0-24 months in low- and middle-income countries: A systematic review protocol” for publication in PLOS ONE. We appreciate the time and effort that you and the reviewers dedicated to providing feedback on our manuscript and are grateful for the insightful comments on and valuable improvements to our paper. We have incorporated most of the suggestions made by the reviewers. Those changes appear as tracked changes within the manuscript. Please see below a point-by-point response to the reviewers’ comments and concerns. All page numbers refer to the revised manuscript file with tracked changes. 

Reviewers’ suggestions and comments to the Authors: 

Reviewer #1: Reviewer comments:

1. This is an intriguing and novel submission, addressing a gap in the literature. The outcomes of the meta-analysis will be valuable in contributing to ongoing efforts to reduce diaper dermatitis. However, I recommend reconsidering the paragraph on page 5, beginning with "Diaper dermatitis...," as it does not substantially enhance the background relevant to the study and could be omitted. Additionally, there is insufficient discussion regarding low- and middle-income countries within the introduction portion of the paper. It would be beneficial to expand on the significance of these populations and clarify whether comparisons will be made to developed countries. Finally, the overarching aim of the review appears somewhat simplistic, and further clarification on how the results will guide future research in diaper dermatitis would strengthen the manuscript.

Response: Thank you for your valuable input, and we appreciate your observation. The introduction section on pages 4, 5, and 6 has been revised to offer a clearer explanation of the subject (diaper dermatitis). We have attempted to investigate the prevalence and contributing variables of diaper dermatitis (DD) in low- and middle-income countries (LMICs), despite the dearth of literature in the target population. The significance of LMICs populations and any potential implications of the study's findings were added in the introduction section.

Reviewer #2: Reviewer comments:

1. Overall: 

It was not clear to me how “low- and middle-income countries (LMIC) will be defined. What criterion will be used?

Response: Thank you for the insightful feedback. To determine a country's economic standing, we shall employ the World Bank's economic classification system. Based on Gross National Income (GNI) per capita, the World Bank divides the world's economies into four income groups: low, lower-middle, upper-middle, and high income. A country is classified as low-income (LIC) if its gross national product (GNI) per person is $1,145 or less; as lower middle-income (LMIC) if it is between $1,146 and $4,515; and as upper middle-income (UMIC) if it is between $4,516 and $14,005. As a result, low, lower-middle, and upper-middle income nations are included in this study. 

2. The title of the protocol states that it will be looking at Diaper Dermatitis “and associated factors”, and the protocol states a key question as “2) What are the factors associated with diaper dermatitis among children aged 0-24 months in low- and middle-income countries?” It is not clear if this includes “causes” of diaper dermatitis, sequelae of diaper dermatitis (for example, ill temperament of the child, poor sleep, pain/crying) or other factors.

Response: Thank you for pointing this out. Our second objective is to identify factors that are associated with DD. Associated factors are correlated variables that may not necessarily have a causal relationship with the outcome variable, such as the child's age, household income, type of diaper used, feeding habits, and use of diaper barriers. These factors can provide valuable information about potential factors that may be related to diaper dermatitis.

3. There is a statement in the “Introduction” related to the prevalence of diaper dermatitis. “Although the prevalence of DD varies from country to country, it is estimated that about 50% to 65% of infants and young children will suffer from diaper dermatitis at some time and up to 25% of children seek healthcare due to this problem (6, 7, 8).” The literature is quite scattered about the percentage of babies who experience diaper dermatitis. However, data from Visscher et al (Pediatric Dermatology Vol 17, No. 52-57, 2000) , indicates that diaper dermatitis rates are ~ 70% in the first week of life which is higher than the 50-65% reported in the publication. Further, Gustin et al (Pediatric Dermatology 2021. Jul;38(4):768-774) indicated that diaper dermatitis was found in 65-83% of newborns. Carr et al. (Pediatric Dermatology 2020 Jan;37(1):130-136) also found high rates of diaper dermatitis in Germany across 4 common sites within the diaper. Given these data, it should be noted that in reality, ‘every baby will experience diaper dermatitis as some point during their diaper wearing years’.

Response: we are grateful for your valuable feedback. We have revised the mentioned points. Prevalence of diaper dermatitis varies by country and study year. We made an effort to use recent sources, for we don't believe that data from 30 or 40 years ago can accurately represent the prevalence of DD at this time. This is the reason why some research article findings were not reported. On the other hand, we will extract data from any research article reporting the prevalence of DD without a time limit.

4. The Introduction also states “Its typical sign is a red, itchy, and sometimes ulcerated rash in the diaper area (perineum, peri-genital area, perianal area, and the upper thigh) occurring most often as a reaction to increased moisture due to prolonged contact with urine, feces, or retained soap or detergent (14).” I would change this to read: “It’s typical sign is a red, itchy, and sometimes ulcerated rash in the diaper area and may include the presence of papules and pustules (Buckley et al, Pediatric Dermatology Vol 33 No 6 632-639, 2016; Odio MR et al Dermatology 2000; 200:238-243). The primary diaper sites include the perineum, peri-genital area, perianal area, genital region, buttocks and groin and occur most often as a reaction to increased moisture (skin overhydration) due to prolonged contact with urine, feces, or retained soap or detergent (14).”

Response: Thank you for your valuable suggestion. We have accepted and amended accordingly.

5. Introduction: “Major causes that impair the integrity of the skin are irritants in urine and feces, overhydration of the epidermal stratum and corneum, excessive skin pH, and friction against the skin (14-16).” I would reword this to say “…elevated skin pH…”

Response: We appreciate your suggestion. We have accepted it. 

6. Introduction: “Disposable diapers contain absorbent chemicals that are able to absorb liquids, and have to be disposed of after usage.”. I would rephrase to state “Disposable diapers contain absorbent gel materials (AGM) that are able to absorb liquid and runny feces and are single-use.”

Response: Thank you for the suggestion. We have accepted it.

7. The Search Strategy is listed as: “The search strategy included a combination of subject terms and free text terms combined with Boolean operators (‘OR’, and ‘AND’). The Medical Subject Headings (MeSH) terms included “diaper”, OR “napkin”, OR “nappy” AND “dermatitis”, OR “rash”, OR “irritation”, AND “neonate”, OR “infant”, OR “toddler”, OR “children”, OR “pediatrics”, OR “kids”, OR “babies”, AND “developing countries”, OR “resource-limit” OR “resource-poor”, OR “low-income”, OR

“lower-middle-income”, OR “middle-upper income countries”.” I would also include “papule”, “pustule”, . I believe search terms for “income” can be included but I strongly suggest the investigators define the criterion for LMIC and then include only those publications where the study was conducted in a LMIC, as a publication may not identify its sources as LMIC.

Response: Thank you for the suggestion. We have included “papule”, “pustule” and “income” terms in our search strategy. Regarding LMIC, the World Bank publishes every country’s economic class. So that the authors are responsible for referring that classification in order to include in the review. We have also included “Operational definition” sub-heading on the methods section on Page 11 to clarify low- and middle-income countries (LMIC). If required, we can attach the list of countries under LMIC economic classes.

8. The inclusion criteria includes data collected at the “under-five clinic”, however, given the episodic, unpredictable and short-term nature of diaper dermatitis (typically 3-10 days) the incidence of diaper dermatitis that is observed at a clinic visit may not reflect the actual rate of diaper dermatitis in the population.

Response: We appreciate your feedback. We also believe that the prevalence of diaper dermatitis at health institutions (clinics) and in the community is not the same. However, our objective is to estimate the pooled prevalence of DD in the health institution among children who come for immunization and other services. For our study is a systematic review, we chose it following a review of literature. After conducting a preliminary search, we found that the majority of the research articles reporting the prevalence of DD were carried out in health facilities. As a result, the study may not represent the prevalence of DD. The study may therefore not accurately reflect the prevalence of DD at the community level. The issue has been recognized as a limitation of the study.

9. The “Search date” is listed as “All research articles accessed from 20 September 2024 to 10 October 2024 without time limits will be included.”. This date is in the past.

Response: Thank you for pointing this out. Our plan was commencing the literature search, and dealing with the protocol simultaneously. However, we have accepted the comment and updated the search date to 10 to 30 November 2024.

10. In the exclusion criteria it states, “Studies that did not assess the magnitude of diaper dermatitis/rashes (studies that evaluate treatments for children with DD, knowledge of DD, the quality of life of children with DD, etc.) will be excluded.”. If parental habits and practices are able to be obtained, this will be critical to understand the potential causes of the diaper dermatitis, as are skin products being used, bathing frequency, diet information and family history of skin disorders, including atopic dermatitis and the like.

Response: We value your opinion. Estimating the pooled prevalence of DD and identifying risk variables for DD are the two main objectives of this review. Any study that reports prevalence of dermatitis/rashes and is conducted in LMIC will be included. On the other hand, if the study doesn’t report prevalence of DD (i.e., their objective is not to report prevalence of DD), they are out of our scope. Therefore, as it is not the objective of the current study, it would not be relevant to include those studies.

---

## [Decision Letter · Decision Letter 1]

4 Nov 2024

Diaper dermatitis and associated factors among children aged 0-24 months in low- and middle-income countries: A systematic review protocol

PONE-D-24-40961R1

Dear Dr. Negera,

We’re pleased to inform you that your manuscript has been judged scientifically suitable for publication and will be formally accepted for publication once it meets all outstanding technical requirements.

Kind regards,

Alison Parker

Academic Editor

PLOS ONE

Additional Editor Comments (optional):

Reviewers' comments:

Reviewer's Responses to Questions

**Comments to the Author**

1. Does the manuscript provide a valid rationale for the proposed study, with clearly identified and justified research questions?

Reviewer #2: Yes

2. Is the protocol technically sound and planned in a manner that will lead to a meaningful outcome and allow testing the stated hypotheses?

Reviewer #2: Yes

3. Is the methodology feasible and described in sufficient detail to allow the work to be replicable?

Reviewer #2: Yes

4. Have the authors described where all data underlying the findings will be made available when the study is complete?

Reviewer #2: Yes

5. Is the manuscript presented in an intelligible fashion and written in standard English?

Reviewer #2: Yes

6. Review Comments to the Author

You may also provide optional suggestions and comments to authors that they might find helpful in planning their study.

Reviewer #2: I thank the authors for the numerous clarifications to the protocol. Best wishes for successful findings.

7. PLOS authors have the option to publish the peer review history of their article (what does this mean?). If published, this will include your full peer review and any attached files.

Reviewer #2: No

---

## [Editor Report · Acceptance letter]

7 Nov 2024

PONE-D-24-40961R1 

PLOS ONE

Dear Dr. Negera, 

I'm pleased to inform you that your manuscript has been deemed suitable for publication in PLOS ONE. Congratulations! Your manuscript is now being handed over to our production team.

Kind regards, 

on behalf of

Dr. Alison Parker 

Academic Editor

PLOS ONE